# Proposed Machine Learning Techniques for Bridge Structural Health Monitoring: A Laboratory Study

**Azadeh Noori Hoshyar** [1,*]**, Maria Rashidi** [2]**, Yang Yu** [3] **and Bijan Samali** [2]

1    Institute of Innovation, Science and Sustainability, Federation University Australia,
     Brisbane, QLD 4001, Australia
2    Centre for Infrastructure Engineering, School of Engineering, Design and Built Environment,
     Western Sydney University, Sydney, NSW 2747, Australia
3    Centre for Infrastructure Engineering and Safety, University of New South Wales,
     Kensington, NSW 2052, Australia
*    Correspondence: a.noorihoshyar@federation.edu.au

**Abstract:** Structural health monitoring for bridges is a crucial concern in engineering due to the degradation risks caused by defects, which can become worse over time. In this respect, enhancement of various models that can discriminate between healthy and non-healthy states of structures have received extensive attention. These models are concerned with implementation algorithms, which operate on the feature sets to quantify the bridge's structural health. The functional correlation between the feature set and the health state of the bridge structure is usually difficult to define. Therefore, the models are derived from machine learning techniques. The use of machine learning approaches provides the possibility of automating the SHM procedure and intelligent damage detection. In this study, we propose four classification algorithms to SHM, which uses the concepts of support vector machine (SVM) algorithm. The laboratory experiment, which intended to validate the results, was performed at Western Sydney University (WSU). The results were compared with the basic SVM to evaluate the performance of proposed algorithms.

**Keywords:** bridge monitoring; feature extraction; feature selection; machine learning; crack detection; anomaly detection; structures





## 1. Introduction

Structural health monitoring (SHM) can potentially provide effective solutions to the continuous assessment of infrastructural health. In the United States, every two years, approximately 600,000 bridges are inspected, scaled and rated according to their condition [1]. According to this rating, released by Federal Highway Administration, fewer than 60% of the national bridges are functioning efficiently [1]. This clearly highlights the need for structural health monitoring. Concrete degradation, steel corrosion, changes in boundary conditions, and the weakening of connections in structures over time are major concerns, which gradually deteriorate the bridge's structural integrity and service capability if not maintained [2].

Currently, the main method of physical and functional assessment of conditions in civil infrastructure is manual, and mostly visual, inspection. This method of inspection is time consuming, expensive, and subjective. Studies have declared that such inspections are not always accurate and involve some errors, as they are highly variable and dependent on the inspectors' proficiency, knowledge and experience, and their emotional state and alertness. Several accidents have been reported that can be related to human error-related insufficient inspections and condition assessments [3,4]. For example, I-35W Highway Bridge in Minneapolis (MA, USA) collapsed in 2007, killing 13 people and injuring a further 145. The National Transportation Safety Board classified this collapse as the consequence of safety issues, such as the lack of proper technology to accurately assess bridge conditions [5].

To prevent further incidents, it is necessary to continuously inspect and assess the condition of civil infrastructure with appropriate techniques, verifying safety and serviceability.

Therefore, putting forward a robust paradigm with respect to the aforementioned safety and economical concerns is a major challenge introduced in this study. Automated condition assessment systems based on machine learning (ML) techniques have been known as one of the technologies that can interpret a large volume of inspection data and detect early-stage structural failure. Despite the vast number of past research studies in this field, few robust methods have been proven to effectively indicate an adverse condition of a structure in service. In addition, many researchers have concentrated on evaluating the application of traditional approaches on different structural experiments, but few studies have been conducted into the enhancement of ML-based methods in SHM. Hence, this is the motivation for this present research.

In this study, we have focused on different paradigms based on machine learning. The study's main goal is to investigate and enhance the performance of ML algorithms for damage detection in SHM based on the concepts of misclassified points, combination of kernels, and by using ensemble classifiers. As per our knowledge, no studies have previously used the first algorithm referenced in this paper in SHM application, and no researcher has investigated the combination of chosen kernels and classifiers of this study. However, we limited this investigation to the field of damage detection studies in civil engineering. Therefore, we generate data through smart aggregate (SA)-based transducers and perform intelligent analysis. Four algorithms are suggested to determine the health state of the structure. The most important contributions of this study are as follows:

- Suggesting four SVM-based algorithms to detect damages in SHM;
- Analyzing and comparing our proposed algorithms with other algorithms using statistical tools;
- Verifying the proposed algorithms' effectiveness in detecting and monitoring flexural cracks in simple concrete beams using mounted smart aggregate transducers;
- Verifying the proposed algorithms in detecting and monitoring flexural cracks in reinforced concrete (RC) beams using mounted smart aggregate transducers.

The remainder of the article is organized as follows: the literature is reviewed in Section 2. The classification concept and the proposed learning algorithms are then presented in Section 3. In Section 4, the performance indicators are described. In Section 5, experimental results are investigated to evaluate the validity of the proposed algorithms. Finally, in Section 6, the article is concluded with discussion of its findings' significance.

## 2. Literature Review

A damage detection procedure can be considered as a pattern recognition (PR) problem [6]. The solution requires a classifier that can discriminate structures as either damaged or intact. Concerning AI techniques, the most commonly used classifiers are based on discriminant analysis, artificial neural network (ANN), k-nearest neighborhood, support vector machine (SVM), support vector regression, adaptive neuro-fuzzy system (ANFIS), fuzzy inference system (FIS) and decision trees. These algorithms have been successfully utilized to solve engineering problems [7–13]. However, the efficiency of the solution depends on both extracted descriptors and a selected classifier. The support vector machine has been extensively accepted as an efficient classifier for detecting damages [14–16]. It also outperforms other techniques in this area of research [17–21].

Radhika et al. presented a damage detection method based on wavelet analysis and machine learning approaches. They used the wavelet to extract the statistical features and performed classification through ANN and SVM. Their results showed the superiority of SVM (66%) over ANN (61%) when compared for accuracy in a damage classification problem [17]. In another study, researchers incorporated SVM with ANN to classify structural modifications in bridges through obtained vibration data [18]. They confirmed the effectiveness of the method in the continuous monitoring of bridges. Bo et al.'s study examined the early examples of intelligence-based algorithms used for detection of structural cracks in

an offshore environment [19]. They used the strain model differences as input parameters, and two artificial intelligence (AI) techniques were used to identify cracks—namely, a support vector machine and neural network. The results show the dramatic changes in strain mode in crack areas, which could be identified through both methods. Moreover, Bo et al. compared the performance of both techniques, and indicated that SVM outperformed NN with fewer errors [19].

Another study. presented the SVM-based approach to identify damages in a long-span arch bridge [20]. The researchers treated the variation ratio of curvature mode as the property to train SVM for identifying the damage level. The obtained results came close to the expected outcome. However, the researchers compared the results with the RBF neural network and verified the precision of the SVM-based method. Satpal et al.'s approach investigated the appropriateness of SVM in beam-like health monitoring through the vibration-induced modal displacement data [22]. The support vector machine was used to predict the damage location and intensity through the displacements of the first mode shape. They simulated 12 different levels of damage intensities with added white Gaussian noise. They also performed 1008 simulations and used SVM to train 90% of data, while 10% was used for testing. Their results showed that SVM has errors varying from 0.28% to 4.57% and 0% to 20.3% in predicting the location and intensity of damages, respectively, without the presence of noise. They also indicated that the presence of noise could decrease the SVM performance. However, they introduced SVM as an effective tool for structural health monitoring. There are many other studies in various domains that report the use of SVM as an effective approach [23–30]. Table 1 summarizes some of other SVM-related studies in SHM, which investigated kernels to develop more effective algorithms.

**Table 1.** Summary of SVM studies based on kernels.

| Ref No. | Algorithm | Domain and Outcome |
|---------|-----------|--------------------|
| [31] | SVM | • Concrete strength<br>• The R-square errors are:<br>• 0.8115 and 0.8227 (radial basis function (RBF) and polynomial kernel function-training)<br>• 0.9422 and 0.9327 (RBF and polynomial kernel function -testing)<br><br>Conclusion: Successful performance of the both the *SVM* and the polynomial kernel. |
| [32] | SVM-combination of kernels (spline and wavelet) | ■ Four-story steel structure<br>■ Enhanced accuracy about (0–8%) than;<br>　(1) Simple wavelet<br>　(2) Gaussian RBF<br>　(3) Thin plate spline RBF<br>　(4) Morlet wavelet<br>　(5) Sinc wavelet<br>　(6) Shannon wavelet<br>　(7) Littlewood–Paley wavelet<br>■ Enhanced accuracy about (2–4%) than;<br>　(8) Gaussian RBF + polynomial<br>　(9) Gaussian RBF + linear<br>　(10) Gaussian RBF + sinc wavelet<br><br>Conclusion: Hybrid kernels can be helpful in enhancing the accuracy, No investigation has been provided on performance of sigmoid kernel alone in this area or combined with any other kernels. Based on our knowledge, no or few investigations have been conducted on the applicability of a sigmoid kernel for damage detection in civil area. |

| Ref No. | Algorithm | Domain and Outcome |
|---|---|---|
| [33] | Combination of kernels (Gaussian RBF and Polynomial) | ■ Better dissemination ability than the Gaussian RBF. Gaussian RBF has 6.8% higher error than the combined kernel. <br><br> Conclusion: Hybrid kernels can be helpful in enhancing the accuracy. No investigation has been provided on performance of polynomial kernel rather than the combined kernels. |
| [34] | Linear, radial, polynomial and sigmoid kernel-based Support vector machine (SVM) | ■ Biomedical engineering <br> ■ Radial- and sigmoid-based SVM outperformed the polynomial and linear kernels significantly ($p < 0.05$) <br><br> Conclusion: Sigmoid kernel has provided better accuracy than other kernels in biomedical engineering. Based on our knowledge, no or few investigations have been conducted on the applicability of sigmoid kernel for damage detection in a civil engineering context. |

This literature review aims to identify the gap in research regarding the algorithms and SHM domain. It firstly indicates that the SVM-based model is superior to the other ML models. Due to its strong theoretical statistical framework, it has proven to be more robust in various applications and when data are integrated with noise. It also possesses high adaptability, good generalization performance, and the capability for global optimization independent of the dimensionality of the input data. SVM provides the least error and shortest processing times [35,36]. The proposed algorithms in this paper are based on SVM models. SVM has successfully solved classification, forecasting, and regression problems. Secondly, despite the diversity of kernels in SVM, the kernel combination is application dependent; however, this hybridization can outperform the single kernels. However, more investigation needs to be conducted into the kernel combinations' performance. The accuracy will enhance if the kernel is aligned with the target information, leading to low errors [37]. Thirdly, ensemble learning is an effective technique that can enhance the models' performance; its accuracy is also generally consistent. This method can also provide a critical boost to different challenges and generate better outputs by combining multiple results [38].

In the following section, the details of the proposed algorithms are explained, preceded by a brief review of fundamental algorithms.

## 3. Learning Algorithms

In this section, we detail the study's key concepts and proposed algorithms.

### 3.1. Support Vector Machine (SVM)

This algorithm [39] was essentially defined to distinguish between two classes, but it could also solve multi-class problems. SVM provided the optimal hyperplane by separating two classes with the maximum margin from the hyperplane. The algorithm assumed an SVM binary classification with N samples in the data training set $(x_1, y_1), \ldots, (x_i, y_i), \ldots,$ $(x_N, y_N) \in R_n \times \{\pm 1\}$, where $x_i$ is a feature vector and $y_i$ is the label of its class. The purpose of SVM was to find the line in such a way that provided the largest minimum distance from the labelled training data, known as the maximum margin hyperplane. Figure 1 demonstrates the basic concept of SVM.

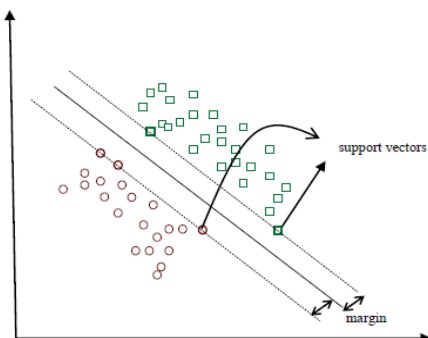

**Figure 1.** Basic concept of SVM [39].

The hyperplane is mathematically defined as a pair (w, b) through < w, x > + b = 0 formula. The hyperplane should meet the following condition, as shown in Equation (1), when linearly separating the train data:

$$y_i(w \times x_i + b) \geq 1, i = 1, \dots, N \tag{1}$$

The distance between the hyperplane and each of the training data $x_i$ is defined through Equation (2) as:

$$d_i = \frac{w \times x_i + b}{\|w\|} \tag{2}$$

By considering both Equations (1) and (2), the following result is obtained for all $x_i$ as:

$$y_i d_i \geq \frac{1}{\|w\|} \tag{3}$$

where the $\frac{1}{\|w\|}$ is considered as the lower distance bound between the hyperplane and the training data $x_i$. The maximum margin of the hyperplane is obtained through minimising Equation (4):

$$z = \frac{1}{2}w \times w \tag{4}$$

Therefore, the final decision function is given as [40]:

$$f(x) = \text{sign}(\sum\nolimits_{\alpha_i > 0} y_i \alpha_i < x, \, x_i > +b) \tag{5}$$

where $(\alpha_1, \alpha_2, \dots, \alpha_N)$ indicates the non-negative Lagrange multipliers related to the constraints, as shown in Equation (1). However, most of these $\alpha$ were usually zero, leading the small proportion of the training data x being considered as vector w. Since these points were considered as the closest points to the hyperplane, they were named the support vectors (SV). SVs were the training patterns which fell on the margin boundaries. In general, a small portion of training sample SVs was used by SVM for the purpose of classification. However, if the SVs consisted of training data beyond the corresponding margins, they were referred to as misclassified data [41]. If the linear separation of data was not feasible, then the hyperplane was pointless. Therefore, N non-negative $(\xi_i, i = 1, \dots, N)$ variables are considered such that:

$$y_i(w \times x_i + b) \geq 1 - \xi_i \tag{6}$$

This variable, $\xi_i$, included a very small quantity of misclassified points and would be zero if the Equation (1) condition was met; otherwise, the term $-\xi$ was added to Equation (1) and generates Equation (6). However, the tolerance parameter could also

cause the ignorance of some training data to make the linear hyperplane. In this situation, the generalized separation hyperplane is expressed by minimising Equation (7):

$$z = \frac{1}{2}w \times w + c\sum_{i=1}^{N}\xi_i \tag{7}$$

The $c\sum_{i=1}^{N}\xi_i$ expression supervised the number of misclassified points. For the smaller value of c, the above solution maximized the minimum distance of $1/w$, and for the large value of c, it minimised the number of misclassified data. It was noted that the use of misclassified data could help to enhance the performance of classifiers.

Furthermore, SVM could generate non-linear decision functions by projecting the training data to the higher dimensional feature space through the non-linear map $\varnothing(x) : \mathcal{R}^n \rightarrow \mathcal{R}^d$. This mapping was performed through the kernels [42]. Kernels execute all the essential operations in the input space through $k(x_i,x_j) = (\varnothing(x),\varnothing(x))$. The $k(x_i,x_j)$ demonstrated the inner product in feature space and had to meet the Mercer's condition [42]. Equation (8) indicates the kernel-based decision function as:

$$f(x) = \text{sign}(\sum_{\alpha_i > 0} y_i \alpha_i k(x, x_i) + b) \tag{8}$$

### 3.2. Kernels

Learning systems used the kernel functions to enhance their computational power. In fact, these functions mapped the data into the feature space (high dimensional space) to linearly separate the data in the new space [43]. However, the performance of kernel-based classification and transformation methods highly depends on the selection of kernel function and its parameters. Therefore, there are numerous approaches to define a suitable kernel function for classification through machine learning algorithms. Kernel design includes two approaches to table design and kernel function design. In table design, the main focus is to create the kernel table and no kernel function design is required. The elements of the kernel table were generated through the training data and optimization function. The other approach was designing the suitable kernel function for the problem [43]. Although selecting the suitable kernel function was extremely important, the different compound had a different performance, were particular problem and application. Therefore, theoretical methods for kernel selection are not completely advisable, except when being evaluated for a particular problem. Table 2 demonstrates different types of kernel functions [43].

**Table 2.** Different types of kernel functions.

| | |
|---|---|
| RBF | $k(x_i x_j) = e^{-\gamma|x_i - x_j|^2}$ |
| Sigmoid | $k(x_i x_j) = \tanh(\gamma x_i^T x_j + c)$ |
| Polynomial | $k(x_i x_j) = (x_i x_j + c)^a$ |
| Wavelet | $k(x, x') = \prod_{i=1}^{n}\left(\cos\left(1.75 \times \frac{x_i - xi'}{\sigma}\right)\exp\left(-\frac{\|x-x'\|^2}{2\sigma^2}\right)\right)$ |
| Chebyshev | $k(x, z) = \sum_{j=0}^{n} U_i(x)U_j^T\sqrt{a - \langle x, z\rangle}$ |
| Gaussian radial basis | $k(x, x') = \exp\left(-\frac{\|x-x'\|^2}{2\sigma^2}\right)$ |
| Exponential radial basis | $k(x, x') = \exp\left(-\frac{\|x-x'\|^2}{2\sigma^2}\right)$ |
| Multi-layer Fourier series | $k(x, x') = \tanh(\rho(x, x') + e)$ |
| Fourier series | $k(x, x') = \frac{\sin\left(N+\frac{1}{2}\right)(x-x')}{\sin\left(\frac{1}{2}(x-x')\right)}$ |
| Splines | $k(x, x') = \sum_{r=0}^{k} x^T x^T + \sum_{r=0}^{k}(x - T_s)_+^k(x' - T_s)_+^k$ |
| B splines | $k(x, x') = B_{2N+1}(x - x')$ |

It is important to address the challenge of making the appropriate kernel functions, as the proper selection or construction of kernel functions can significantly affect the performance of learning systems.

### 3.3. Proposed Algorithms

The details of the four SVM-based algorithms proposed in the present study are explained in the following.

### 3.3.1. SVM Based on Misclassified Data (SVM-MD)

There has been a substantial increase in the utilization of advice sets in learning algorithms. However, difficulties remain regarding the application and expression of this knowledge in terms of its constraints. Moreover, these techniques require new parameters that can enhance the SVM computational cost. In this regard, this study suggested the non-iterative algorithm, in which the subsequent knowledge is extracted from the training phase to improve the performance of SVM. In the implementation of a basic SVM algorithm, the first type of support vectors or hyperplane position is the only information that is utilized in the test phase of SVM from the training phase; the purpose of this algorithm is utilizing the second type of support vectors to provide the subsequent knowledge by the purpose of enhancement in SVM performance. The second type of support vectors was selected, as there was a lot of misclassified data, even in the existence of the optimized hyperplane [41]. Two potential sources of this misclassified data were the outliers and the non-linear separable data when using kernels. Basic SVMs do not consider non-linear separable data during the training phase. This happens in defining the constraints and the tolerance parameters of objective functions. The main concern herein is that, if the data in the test set appear practically the same as these misclassified data in the training set, they would be classified incorrectly. This misclassification could reduce accuracy, as it happened because the SVM ignored the information in the training phase. Therefore, to obtain benefits from misclassified data, the outlier's effect should be taken into account. Searching to find more similar data to the misclassified data, as proposed in this research, could enhance SVM performance. In this method, after determining the misclassified data in the training phase, the SVM provided the advice weights which were used in conjunction with the decision values in the testing phase. These defined advised weights helped SVM to improve its accuracy while eliminating the outlier data. Equation (9) expresses the obtained misclassified data of the training phase as [44]:

$$MD = \bigcup_{i=1}^{N} x_i | y_i \neq \text{sign}(\sum_{\alpha_j > 0} y_j \alpha_j k(x_i, x_j) + b) \tag{9}$$

The right side of the above equation may include any SVM decision function. Although there was the possibility that misclassified data be null, experimental results showed that the incident of misclassified data was prevalent. For each misclassified data in $M_d$ set, the length from the corresponding k-nearest neighbour (k = 10), which have been correctly classified, is computed as:

$$CL(x_i) = \text{Minimum}_{x_j}(\|cl(x_i) - cl(x_j)\|) \tag{10}$$

$$cl(x_i) = \frac{x_i}{M_d} \tag{11}$$

$$cl(x_j) = \frac{x_j}{M_d} \tag{12}$$

$$M_d = \frac{1}{N}\sum_{j=1}^{N} (x_i - x_j) \cdot N = 1, 2, \ldots, 10, \ y_i \# y_j \tag{13}$$

where $CL(x_i)$ is the disperse of misclassified data from the 10-nearest neighbour which has been correctly classified. $M_d$ represents the average of all distances within the k-nearest neighbour (k = 10) of $x_i$. However, by mapping the training data into higher dimension, Equation (14) can be used with respect to kernel k to compute the length as:

$$(\|\theta cl(x_i) - \theta cl(x_j))\|) = (k(cl(x_i), cl(x_i))) + (k(cl(x_j), cl(x_j))) - 2k(cl(x_i), cl(x_j))0.5 \tag{14}$$

By computing the CL for each of the $x_k$ in the test set, the self-weighting (SW) is assigned as:

$$SW = \begin{cases} 0 \; \forall x_i \in MD, \; \text{if MD is empty or} (||cl(x_k) - cl(x_i)||) > CL(x_i) \\ 1 - \frac{\sum_{xi} ||cl(x_k) - cl(x_i)||}{\sum_{xi} CL(x_i)} x_i \in MD, \; \text{if MD is empty or} (||cl(x_k) - cl(x_i)||) \leq CL(x_i) \end{cases} \quad (15)$$

The obtained SWs show the closeness of the test data to the misclassified data in the training phase. Therefore, the processing flow of proposed self-weighting SVM could be written as:

- In training phase, perform the SVM training;
- Use Equation (9) to find the misclassified data (MD);
- Investigate the existence of misclassified data maintaining in a MD structure. If the MD includes the data, the CL is computed through Equation (10) for each member of MD; otherwise, the normal procedure of SVM is continued;
- In the testing phase, compute the self-advised weights of each $x_k$ in the test set;
- For each $x_k$ in the test set, the absolute values of SVM decision values are computed and scaled to [0, 1];
- The SVM labelling is followed based on the conditions in Equation (15). The normal SVM labelling is performed if SW $(x_k)$ < decision value $(x_k)$.

However, this algorithm required a large amount of memory to store all of the support vectors and was not suitable for datasets with a large number of features, as it could become computationally intractable. On the other hand, it could be used in a wide range of applications, from text classification to image classification.

### 3.3.2. SVM Based on Hybrid Kernels

Although the concept of hybrid kernels was not new, limited prior research had been performed in in that area. The main concern of this section was to enhance the models based on new hybrid kernels that outperformed SVM with traditional kernels. As illustrated in various studies [27,45], generating the new kernel functions through the use of existing kernel functions was more efficient. Therefore, in this section, two hybrid kernel functions were generated through the use of polynomial and sigmoid kernels. The proposed hybrid kernels enhanced the accuracy, while keeping the number of support vectors in the range of required support vectors. In this method, the operators were applied on multiple kernel functions to provide new kernel functions, which meet the properties of each kernel combined. These operations directly affected the kernel matrix and the operation result was the positive semi-definite matrix at all times. The polynomial kernel function was a global kernel function that provided a better dissemination capability and a weaker learning ability [33], while the sigmoid kernel function provided a better global performance [46].

$$\text{Sigmoid } K(x_i x_j) = \tanh\left(\gamma x_i^T x_j + c\right) \quad (16)$$

$$\text{Polynomial } K(x_i x_j) = (x_i x_j + c)^a \quad (17)$$

To provide a model that employed the advantages of both kernels, new kernels were generated based on $K_{M1}$ and $K_S$, which could provide better dissemination capability and better generalization ability. The new kernels can be formed as:

$$K_{M1}(x, z) = k_1(x, z)k_1(x, z) \quad (18)$$

$$K_{M2}(x, z) = k_2(x, z)k_2(x, z) \quad (19)$$

$$K_S(x, z) = \alpha k_{M1}(x, z) + \alpha k_{M2}(x, z) \tag{20}$$

where $k_1$ is the sigmoid kernel function with slope gamma and intercept c, $k_2$ is the polynomial kernel over $X \times X \in R_n$, $\alpha \epsilon R+$. Therefore, the new kernel functions are mathematically expressed as:

$$K_{M1} = \tan h(\gamma x_i^T x_j + c) * \tan h(\gamma x_i^T x_j + c) \tag{21}$$

$$K_{M2} = (x_i x_j + c)^a * (x_i x_j + c)^a \tag{22}$$

$$K_S = \alpha(\tan h(\gamma x_i^T x_j + c) * \tan h(\gamma x_i^T x_j + c)) + \alpha((x_i x_j + c)^a * (x_i x_j + c)^a) \tag{23}$$

In mathematics, a proof of Equation (20) to determine that "the sum of two kernel functions still a kernel function" is shown as follows. The equation assumed that S is a set of finite points $\{x_1, x_2, \ldots, x_n\}$, while $k_1$ and $k_2$ were the corresponding kernels resulted from the $k_1$ and $k_2$ on the restricted points, respectively. As illustrated above, k is the positive semi-definite matrix for all $\alpha \epsilon R+$ as:

$$\alpha' k \alpha \geq 0 \tag{24}$$

then,

$$\alpha'(k_1 + k_2)\alpha = \alpha' k_1 \alpha + \alpha' k_2 \alpha \geq 0 \tag{25}$$

Therefore, $k_1 + k_2 \geq 0$ is the positive semi-definite matrix and still a kernel function.

The proposed SVMs, which are based on $K_{M1}$ and $K_S$ kernels, are called SVM-S2 and SVM-SP, respectively.

Therefore, the processing flow of the enhanced algorithm can be written as:

---

**Algorithm 1** SVM-S2 and SVM-SP

---

1. Implement classic SVM.
2. Tuning SVM parameters.
3. Set the kernel function.

If SVM-S2:

Set the kernel function to Equation (21).

If SVM-SP:

Set the kernel function to Equation (23).

4. SVM training process.
5. K-fold cross validation.
6. SVM forecasting process.

Compute the accuracy (Acc1) and F-score (F1-S).

---

### 3.3.3. SVM Based on Ensemble Classifiers (SVM-EN)

Ensemble classifiers were the machine learning algorithms that utilized multiple learning algorithms to enhance the predictive performance that could be achieved by learning algorithms alone. The hybrid classifiers, therefore, provided better performance and accuracy than the simple individual classifier [47–49]. However, to build an efficient hybrid classifier system, it was required to choose the generation model scheme as well as the classification methods. There are many schemes that perform the model's combination [48,50,51]. The main challenge that arose here was choosing and combining the diverse methods and models. The reliability and effectiveness of the system could have been directly affected by these selections, depending on the application. Therefore, the appropriate combination strategy reduced the prognosis errors.

Figures 2 and 3 show two types of approaches in the model generating paradigm [52]. Figure 2 is a homogeneous model [53,54], which generates the model by employing a single learning algorithm to the different partitions of the feature vectors in the dataset.

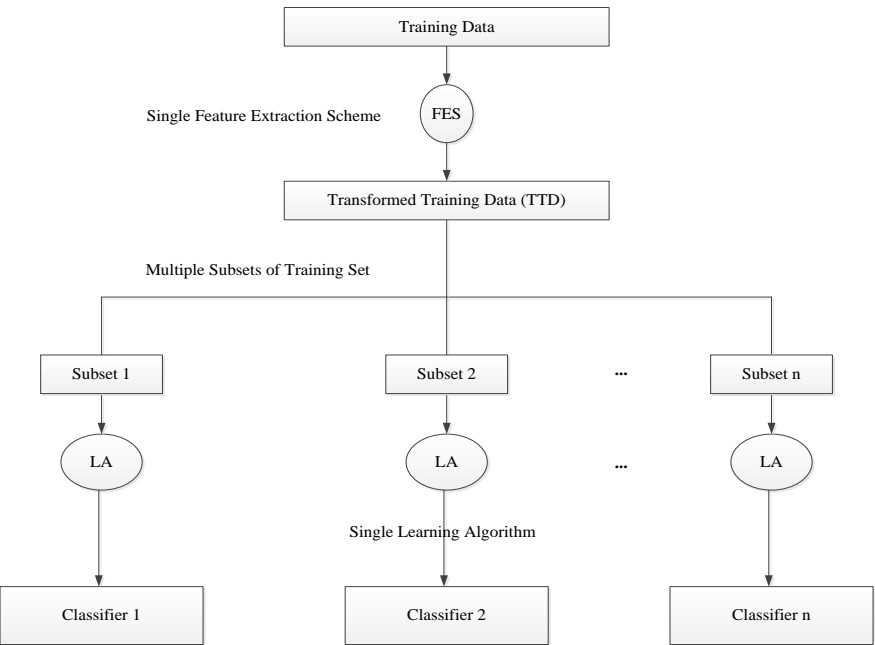

**Figure 2.** Homogeneous paradigm.

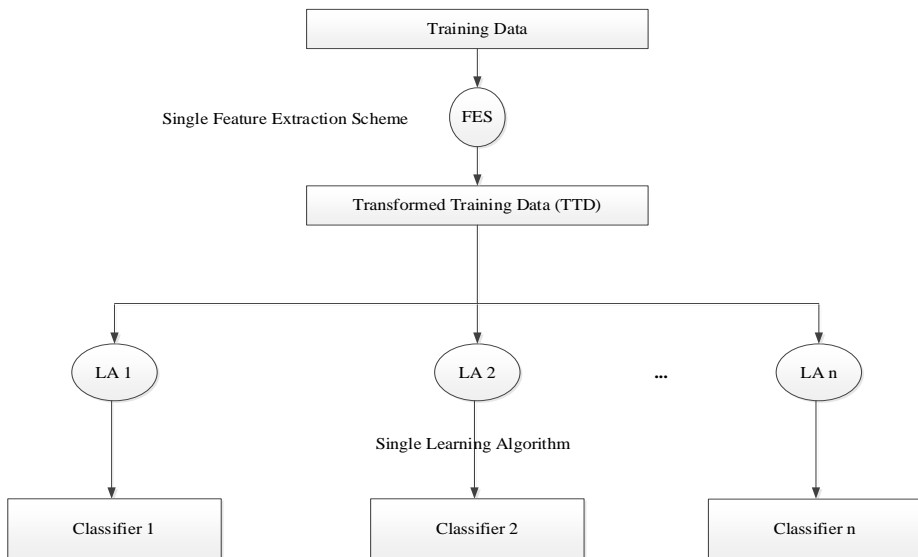

**Figure 3.** Heterogeneous paradigm.

Figure 3 is a heterogeneous model [52], which generates models by employing various learning algorithms with the same dataset of feature vectors. The obtained outcome of classifiers could be combined through different techniques, such as cascading, voting, and stacking. It should be noted that the ensemble models can solve various potential issues by only using the single classifier. These issues were associated with the proper model selection, choosing the correct local minimum and the infeasibility of the search space expansion [55].

The present study followed the heterogeneous paradigm, as seen in Figure 3. Therefore, the aggregation of the results of multiple classifiers with the ultimate goal of accuracy

enhancement was presented. This ensemble embraced a set of individually trained classifiers: the enhanced support vector machine (I), the enhanced support vector machine (II), and the k-nearest neighbour classification model [56]. The process of ensemble design in this study followed two main steps. The first step was the model training, in which the above models were trained with the same training set. The second step was the model combination, in which the outputs of all classifiers made the final result through the majority voting. The idea of majority voting [57] is that each classifier provides its vote on the specific class; the majority of votes are then considered as the final output. The processing flow of this algorithm is detailed as follows:

---

**Algorithm 2** SVM-EN

---

*Input*: X: training data, Y: class labels of X, K: number of nearest neighbors.
*Output*: Class of a test sample x.
Start
1. Implement algorithm 1, Section 3.3.1.
2. Compute the accuracy (Acc1) and F-score (F1-S).
3. Specify class/label.
4. Implement algorithm 2, Section 3.3.2.
5. Compute the accuracy (Acc2) and F-score (F2-S)
6. Specify class/label.
7. Classify (X, Y, x) by implementing KNN.
*7.1. **For** each sample x do.*
$\qquad$ Calculate the distance: $d(x, X) = \sqrt{\sum_{i=1}^{n}(x_i - X_i)^2}$.
$\qquad$ ***End for.***
$\qquad$ Classify x in the majority class: $C(x_i) = \text{argmax}_k \sum_{x_j \in KNN} C(X_j, Y_k)$.
7.2. Compute the accuracy (Acc3) and F-score (F1-S3).
7.3. Specify class/label.
8. Use majority voting to specify final output based on steps 3, 6, 7.4.
End.

---

## 4. Performance Indications

This study used the confusion matrix, Table 3, to evaluate the performance of algorithms by computing the accuracy and F1-score [57].

$$\text{Accuracy} = \frac{t_p + t_n}{t_p + f_p + f_n + t_n} \qquad (26)$$

$$F_1 - \text{Score} = \frac{2 \times (\text{Recall} \times \text{Precision})}{\text{Recall} \times \text{Precision}} \qquad (27)$$

$$\text{Recall} = \frac{t_p}{t_p + f_n} \qquad (28)$$

$$\text{Precision} = \frac{t_p}{t_p + f_p} \qquad (29)$$

**Table 3.** Confusion matrix.

|  | **Predicted Positive** | **Predicted Negative** |
|---|---|---|
| Label positive | $t_p$: true positive | $f_n$: false negative |
| Label negative | $f_p$: false positive | $t_n$: true negative |

## 5. Experimental Analysis

The main purpose of this section is to present experimental data analysis based on the proposed algorithms. This experiment was performed at Western Sydney University (WSU). MATLAB R2018b software was used to analyze the data.

### 5.1. Concrete Beam Preparation

The blend proportion of the ready-mixed concrete was utilized to cast the concrete beams. The aggregates with the maximum size of 10 mm, a slump of 70 mm, 28 day-compressive strength of 40 MPa, and a water-to-cement ratio of 0.48 were used based on the design created in prior research studies [58]. The Australian Portland cement type GB was utilized and the concrete and sand conformed to AS3600 [58]. The material properties are summarized in Table 4 [58].

**Table 4.** Material properties in concrete mix.

| Materials | Characteristics | Values |
|---|---|---|
| Portland cement type GB | Specific gravity | 3.15 |
| Natural river sand (Fine aggregate) | Specific gravity | 2.55 |
| | Size | 0.15 to 4.75 mm |
| Natural river gravel (Coarse aggregate) | Specific gravity | 2.60 |
| | Maximum size | 10 mm |
| Tap water | Density | 998–1000 kg/m$^3$ |

To test the design requirements, a slum test was carried out before the concrete casting. The standard cylinders were cast with the size of 102 mm × 203 mm. The same environmental conditions were applied to cast and cure the cylinders [58]. Ten concrete beams were tested. Plywood was used to fabricate the molds. After casting, the specimens were vibrated and the surface finishing was performed by hand floating.

### 5.2. Data Collection and Measurement Setup

In this study, the data was collected through the active sensing approach, which measures the propagation of stress waves characterized in concrete.

Figure 4 shows our measurement setup for this study. From a hardware perspective, the PC and SA transducers were connected to the data acquisition board. The guided stress wave was generated in the concrete through the SA actuator and was partially received by the SA sensors. Mounted SA transducers, a newly developed arrangement [58], were used to detect and localize the damage on concrete and reinforced concrete specimens under loading. The distances of 20 mm and 40 mm were selected for SAs and the time-domain signal was recorded.

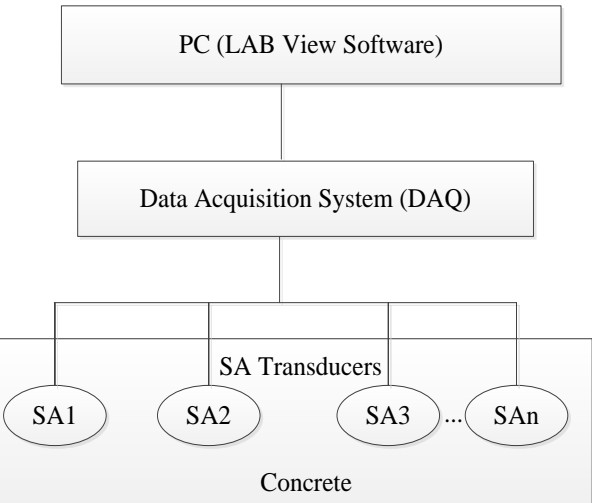

**Figure 4.** Measurement system.

Figures 5 and 6 show SA arrangements for three-point and four-point bending tests on concrete and RC beams at Center of Infrastructure Engineering (CIE), Western Sydney University. The difference between these two experiments is that, in three-point bending test, only one crack appeared in each beam, as the concrete beams were not reinforced. When the crack happened, the concrete beam lost its resistance instantly. In four-point bending test, the crack spread into the neutral axis location, and the expected cracks occurred at the top of the specimen due to the compression effect, which caused the varying transmission properties. However, after loading, multiple foreseen cracks become visible in the mid-span region of the beam due to the concrete beam reinforcement.

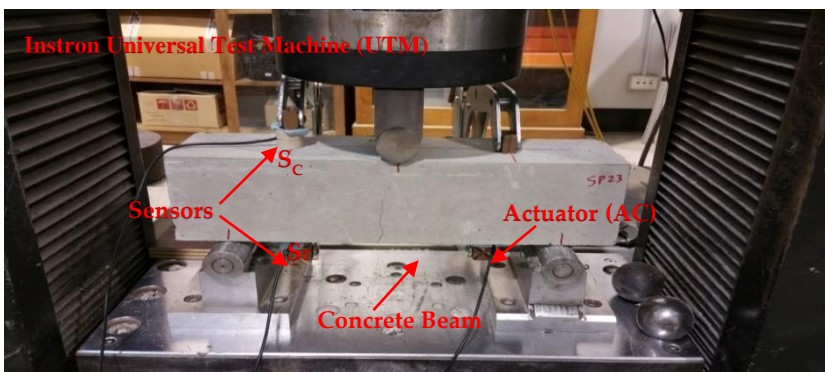

**Figure 5.** The three-point bending test, Center of Infrastructure Engineering (CIE), Western Sydney University.

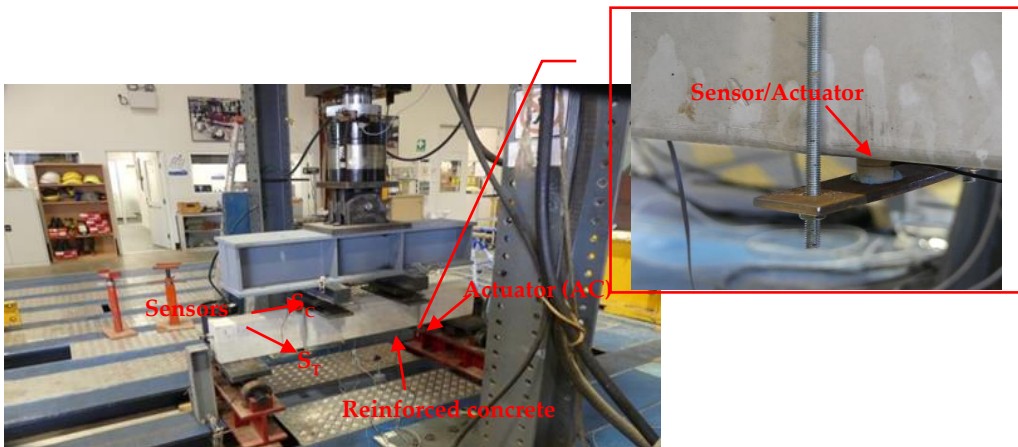

**Figure 6.** Reinforced concrete under four-point bending, Center of Infrastructure Engineering (CIE), Western Sydney University.

This paper employs MATLAB software for analytical parts and LabVIEW software for two purposes. Firstly, the software generates a swept sine wave which is considered as an excitation wave; secondly, the received signals through the SA sensors are processed through the program. The sinusoidal sweep signal of this experiment was generated through the actuator, ranged from 100 Hz to 150 kHz, and has a magnitude of 10 V. The sweep and recording periods are 1 s and 4 s, respectively. [58]. Therefore, during each measurement, at least three complete sweep periods were recorded. However, the continuous signal recording of transducer readings was performed by the rate of 10 channels/s through the automatic data-logger for about 60 min. Ten concrete beams (400 mm × 100 mm × 100 mm) and four RC beams (1700 × 150 × 250 mm) were tested. The test was carried out through the Instron universal test machine (UTM) with a loading capacity of 200 kN and 1000 kN for concrete and RC beams, respectively. The load changes on simple and RC beam specimens were recorded through the software. The loading cell

and displacement movement were set to 0 and 0.01 mm/min for concrete beams and 0 and 0.009 mm/s for RC beams, respectively. This data was used for training and testing the proposed methods. The data logger was calibrated before the beginning of each test.

The received signals were saved in the format of technical data management streaming (TDMS) and extracted to .txt file for further analysis. Figure 7 shows the time-domain signal plotted in LabVIEW.

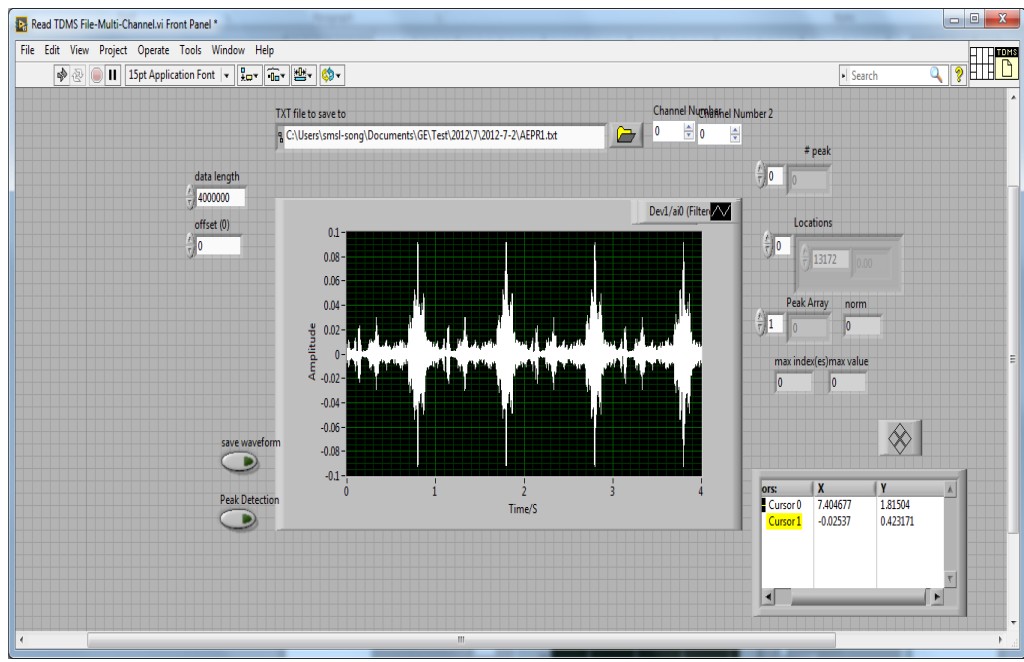

**Figure 7.** TDMS file run in LABVIEW software.

The purpose of this data recording is to monitor the health state of structures in real-time. The continued recording of this data allowed ML techniques to detect concrete failure due to the changes happen in the value of extracted features. In total, 698 signals were recorded.

### 5.3. Feature Extraction

Feature extraction is the second step in every automatic system. In SHM applications, this step extracts the damage indicative features from the raw signals, which represent the most considerable changes in fault occurrence. The main task of feature extraction is to identify the signal characteristics as good indicators of faults in the structure [59].

The main concern regarding feature extraction is missing information, as some knowledge may not be captured through one particular feature or should be classified as noise instead of a feature. Therefore, it is necessary to establish some complementarity features that can represent the signal more accurately; the main challenge of this section is to provide the optimal set of properties for a feature vector. The proper selection of the properties, to provide the highest accuracy in one set, is a high priority for researchers. The combination of several different properties can provide a closer representation of the underlying phenomenon; it is more difficult to establish this understanding with only one property.

In general, the performance of feature sets is relative and can provide different outcomes based on the application. The properties can be extracted in time-domain, frequency domain and time–frequency domain formats. In many signals, the frequency component contains some important information, which is hidden and helpful in distinguishing the patterns in data analysis.

According to the literature [60–63], the following features (shown in Table 5) were extracted from the signals for further processing:

**Table 5.** Extracted features for signal processing.

| | |
|---|---|
| Crest factor: The L infinity norm and RMS values are computed throughout the specified dimension. | $CF = \dfrac{\|X\|_\infty}{\sqrt{\frac{1}{N}\sum_{n=1}^{N}|X_n|^2}},$ |
| Root-mean-square level: where x is a vector, y indicates that the y is a real-valued scalar. | $x_{RMS} = \sqrt{\frac{1}{N}\sum_{n=1}^{N}|x_n|^2},$ |
| Sparse filtering: where, $f_j^{(i)}$ is the jth feature value for the ith column. | $\text{minimize} \sum_{i=1}^{M} \left\|\hat{f}^{(i)}\right\|_1 = \sum_{i=1}^{M} \left\|\dfrac{\tilde{f}^{(i)}}{\left\|\tilde{f}^{(i)}\right\|_2}\right\|_1$ |
| Average frequency: n = number of frequency bins in the spectrum; fi = frequency of spectrum at bin i of n; $I_i$ = Intensity (dB scale) of spectrum at bin i of n. | $f_{mean} = \dfrac{\sum_{i=0}^{n} I_i \times f_i}{\sum_{i=0}^{n} I_i}$ |
| Energy: Z is the magnitude; Es is signal energy. | $E = \dfrac{E_s}{Z} = \frac{1}{Z}\int_{-\infty}^{\infty}|x(t)|^2 dt$ |
| Maximum-to-minimum difference: | $A_{peaktopeak} = A_{average} \times \pi$ |
| Rise level: | $RL = value\,(R_{wave}) - value\,(Q_{wave})$ |
| Fall time: | $t_f$ = Time lasts for the amplitude of a pulse to fall from a specified value to another specified value. |
| Fall level | $FL = value\,(R_{wave}) - value\,(S_{wave})$ |

The above features are extracted from the sensor data and considered as the feature set. For each feature vector, the healthy and cracked states were labelled as 0 and 1, respectively. In the following section, by training the models through the formed feature set, crack identification is performed when the features' values change over the healthy state. The following section investigates the results of models using the feature set.

*5.4. Classification*

This section discusses the results of classification on the experimental data of a simple and RC beam. The obtained features were used to investigate the performance of proposed classification techniques. The classifier learnt from the training data was used to make a decision when testing data were presented. The performance accuracy was computed through the average of 4-fold cross-validation (the data is split into four groups. Three groups were set as the training and validation data and one group was set as the test data) for 100 runs and performed for 29 observations on 699 feature sets. The average of these 29 observations represents the final accuracy and F-score. For all classifiers, the gamma was set to scale, the tolerance was set to 0.001, and the C parameter was set to 1. The kernel for Basic SVM was set to polynomial.

The first analysis trained the SVM-MD algorithm, which was based on the length of misclassified data from the corresponding 10-nearest neighbour, which were then correctly classified. The kernel type was set to polynomial. The performance of this algorithm was obtained by computing the average accuracy. The other analysis is associated with the SVM-S2 and proposed SVM-SP algorithms, which are based on the hybrid kernels. For these two models, the kernel type is set to $K_{M1}$ and $K_S$, respectively. The SVM-EN was introduced as the ensemble classifier and worked through the hybrid learning algorithms and voting system. In this algorithm, for the k-nearest neighbour model, k was set to 5, the distance metric was set to Euclidean distance and weighting was set to inverse distance weighting. Table 6 shows a comparison of accuracy between the proposed algorithms and the basic SVM. The basic SVM is considered as the benchmark.

According to the above results, the analysis conducted by estimating the performance of SVM-MD obtained an accuracy of 87.22% and F-score of 0.80 for simple beam, compared to an accuracy of 84.72% and F-score of 0.58 for the basic SVM. The analysis was repeated by obtaining the average accuracy of 86.82%, 86.46%, and 87.2% and F-score 0.77, 0.73, and 0.79 for the SVM-S2, SVM-SP and SVM-EN, respectively. However, the SVM-MD

and SVM-EN yielded the highest performances; in contrast, the SVM-SP yielded a weaker performance. For RC beam specimens, the SVM-MD model showed an accuracy of 86.29% and F-score of 0.73, which outperforms the other models. The result of this experiment confirms the results obtained through the experiment on a simple beam.

**Table 6.** Comparison of accuracy between basic SVM, SVM-MD, SVM-S2, SVM-SP, and SVM-EN for simple and RC beams.

| Models | [1] S. Beam | | RC Beam | |
|---|---|---|---|---|
| | Acc (%) | F1-S | Acc (%) | F1-S |
| Basic SVM | 84.72 | 0.58 | 85.38 | 0.63 |
| SVM-MD | 87.22 | 0.80 | 86.29 | 0.73 |
| SVM-S2 | 86.82 | 0.77 | 86 | 0.70 |
| SVM-SP | 86.46 | 0.73 | 85.54 | 0.68 |
| SVM-EN | 87.2 | 0.79 | 86.08 | 0.71 |

[1] S. Beam is contraction of simple beam.

Further analyses of these performances were carried out by computing the *p*-value of the *t*-test to determine the average of accuracies (shown in Tables 7 and 8) for simple and RC beams, respectively. The details of these statistics can be found in Appendix A.

**Table 7.** *t*-test: basic SVM and enhanced SVMs sample for variance for simple beam.

| Models | $P(T \leq t)$ One-Tail | $P(T \leq t)$ Two-Tail |
|---|---|---|
| SVM-MD | $1.75 \times 10^{-3}$ | $3.50 \times 10^{-3}$ |
| SVM-S2 | $7.45 \times 10^{-3}$ | $1.49 \times 10^{-3}$ |
| SVM-SP | $2.25 \times 10^{-3}$ | $4.51 \times 10^{-3}$ |
| SVM-EN | $1.92 \times 10^{-3}$ | $3.85 \times 10^{-3}$ |

**Table 8.** *t*-test: basic SVM and enhanced SVMs sample for variance for RC beam.

| Models | $P(T \leq t)$ One-Tail | $P(T \leq t)$ Two-Tail |
|---|---|---|
| SVM-MD | $2.72 \times 10^{-3}$ | $5.44 \times 10^{-3}$ |
| SVM-S2 | $5.57 \times 10^{-3}$ | $1.11 \times 10^{-3}$ |
| SVM-SP | $1.44 \times 10^{-3}$ | $2.87 \times 10^{-3}$ |
| SVM-EN | $8.35 \times 10^{-3}$ | $1.67 \times 10^{-3}$ |

The *p*-value should be less than 0.05 for the results of the test to be considered statistically significant. This means that if the calculated *p*-value is equal to or greater than 0.05, the two samples are not significantly different. The obtained *p*-values in Table 7 demonstrate that the difference between the accuracies of the basic SVM and the proposed classifiers is statistically significant.

We used the confusion matrix to appraise the performance of models. This matrix computed for the known and predicted labels using the "confusionmat" function, which shows the resulting classification for structural status. The "confusionchart" function is used to carry out plotting. The results of the models are presented in Figure 8 for both simple and RC beams. It shows the summary of prediction results based on the extracted features from recorded signals, Section 5.3. The rows and columns correspond to output/predicted and target/true classes, respectively. The diagonal/off-diagonal cells correspond to observations which have been correctly/incorrectly classified. The cells show the number of observations and associated percentage figures. The right column demonstrates the percentages of all correctly/incorrectly classifications (predicted), while the bottom row shows the percentages of all correctly/incorrectly classifications (true belongings to each class). The bottom right cell demonstrates the overall accuracy.

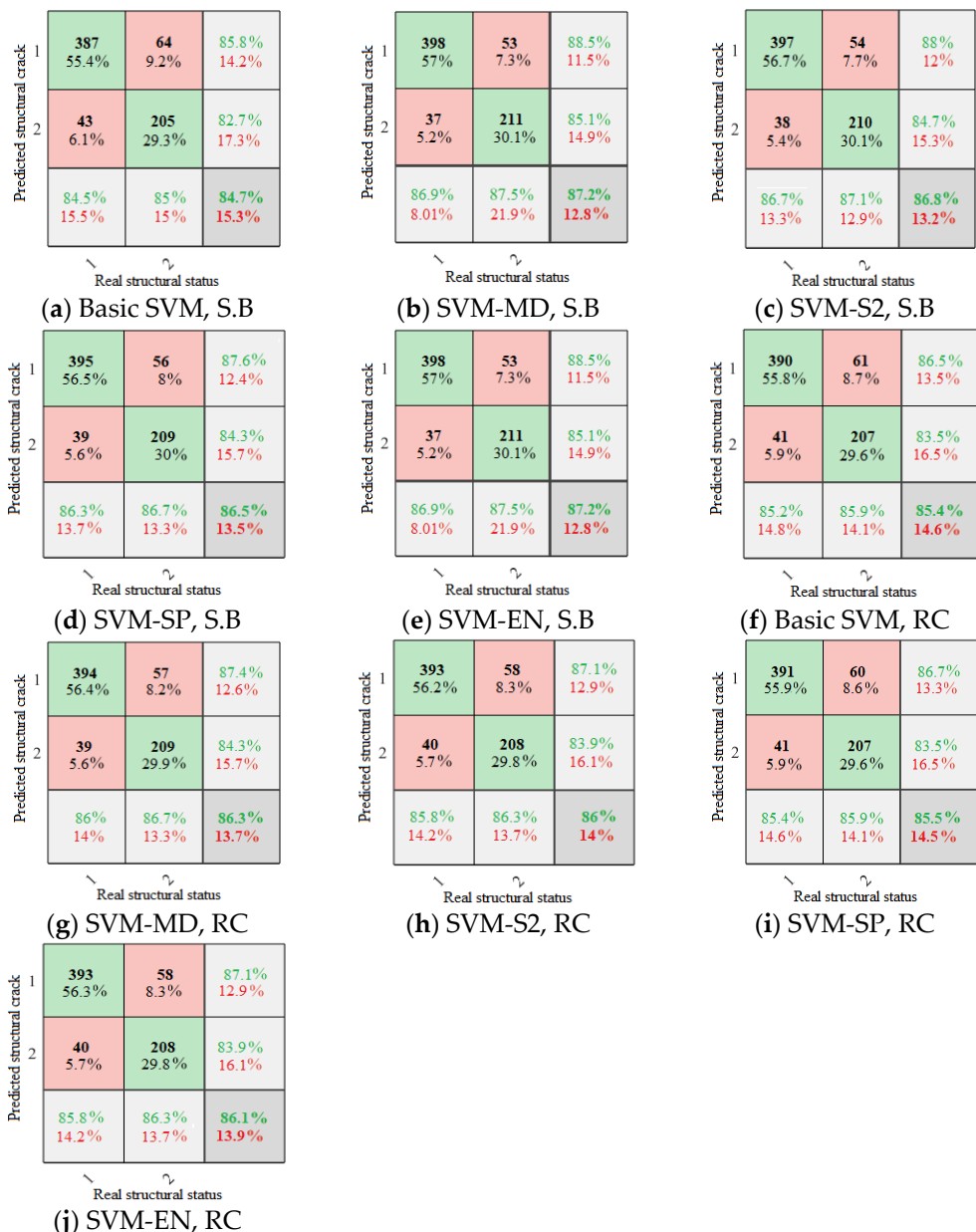

**Figure 8.** Confusion matrix of enhanced SVM models for S.B and RC beams. The rows correspond to the true class, while the columns correspond to the predicted class. Diagonal and off-diagonal cells correspond to correctly and incorrectly classified observations, respectively.

The above figure for confusion matrices shows the number of correctly classified and misclassified data items for each model for both simple and RC beams. In the figure, "b", "c", "d" and "e" correspond to the defect and healthy categories on simple beams using the models. Moreover, "f", "g", "h", "i" and "j" correspond to the defect and healthy categories on RC beams using the models. It can be seen from the results that the prediction accuracies of validation signals by four models are all above 86%.

From these results, its is clear that SVM-MD and SVM-EN in S.B have the optimal performance with overall accuracy of 87.2%, while SVM-SP is the least accurate algorithm in terms of defect classification. For RC beams, SVM-MD outperforms the other algorithms.

For further details on the above results, according to confusion matrix (b), 398 data points known to be in group 1 (healthy structure) are classified correctly. This corresponds to 57% of all 698 signal data. For group 2 (cracked), 53 of the data points are misclassified into group 1. Moreover, 37 of the data points known to be in group 1 are misclassified

into group 2. In comparison with the basic SVM, the number of correctly classified data points has been increased, while the number of misclassifications has been reduced. Similar interpretations are applicable for other confusion matrices, as shown in Figure 8.

To evaluate the performance of models in identifying defects in beams, receiver operator characteristic (ROC) is adopted by plotting true positive rate (TPR) vs. false positive rate (FPR) for SBs. The area under the ROC curve (AUC) is an essential metric for performance assessment in machine learning study. In general, the value of AUC ranges between 0 and 1, where "0" indicates a poor model with the worst performance of separability and "1" indicates the model with excellent measure of separability [64]. As we investigated four models in this study, there are four ROC curves of different colors displayed in the Figure 9, together with corresponding AUC values.

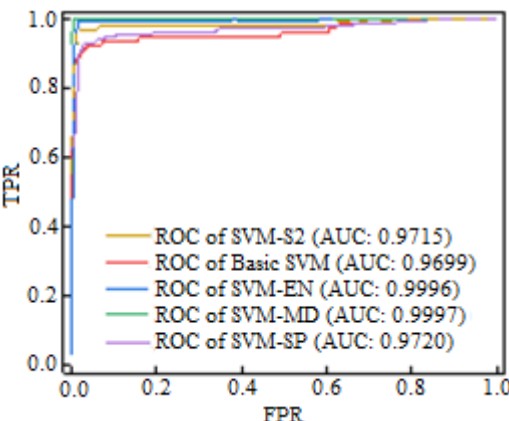

**Figure 9.** ROC curves for the models for defect detection.

Figure 9 displays the ROC curves with AUC values of different models. According to the values, the SVM-MD framework possesses the highest AUC, signifying that it has optimal capacity to detect and classify defects.

## 6. Conclusions

The purpose of this work was to enhance ML-based algorithms for SHM. Four SVM-based approaches were proposed to enhance the performance of SVM for bridge structural health monitoring. In the first approach, additional weights were assigned to the data by computing the length of each misclassified data item from the corresponding k-nearest neighbour (k = 10) which had been correctly classified. The second and third approaches introduced the new kernels through a combination of kernels, as well as polynomial and sigmoid kernel functions. In the fourth approach, the hybrid classifier was proposed through aggregating (majority voting) the different classifiers trained with the same data set. This ensemble embraces a training of individual classifiers, including the SVM-MD, SVM-S2, and the k-nearest neighbour classification model. The results showed the SVM-MD had an accuracy of 87.2% for S.B and 86.3% for RC beams, respectively; it also provided better classification performance than a basic SVM, which had an accuracy of 84.7% for S.B and 85.4% for RC beams, respectively. Though the other algorithms were less accurate than SVM-MD, they also outperformed the basic SVM.

The findings of this paper will firstly help in the health monitoring of bridge structures. The sensors are installed on the structure and the signals received by the sensors are continuously recorded. In the occurrence of a crack, the value of features extracted from the sensor signals differs from the health state. Therefore, the proposed ML-based techniques will classify these variations as crack identification.

Secondly, the paper's findings help researchers to investigate the different models and develop new ones based on their application.

**Author Contributions:** Data curation, M.R.; Writing—original draft, A.N.H.; Writing—review & editing, A.N.H. and Y.Y.; Supervision, B.S. All authors have read and agreed to the published version of the manuscript.

**Funding:** This research received no external funding.

**Informed Consent Statement:** Not applicable.

**Data Availability Statement:** Data is available on request.

**Conflicts of Interest:** The authors declare no conflict of interest.

## Appendix A

**Table A1.** *t*-test: two-sample assuming unequal variances between basic SVM and SVM-MD, simple beam.

| | **Basic SVM** | **SVM-MD** |
|---|---|---|
| Standard deviation | 0.13 | 0.03 |
| Observations | 29 | 29 |
| Hypothesized mean difference | 0 | |
| df | 34 | |
| t Stat | −97.32 | |
| P(T ≤ t) one-tail | $1.75 \times 10^{-3}$ | |
| t Critical one-tail | 1.69 | |
| P(T ≤ t) two-tail | $3.50 \times 10^{-3}$ | |
| t critical two-tail | 2.03 | |

**Table A2.** *t*-test: two-sample assuming unequal variances between basic SVM and SVM-S2, simple beam.

| | **Basic SVM** | **SVM-S2** |
|---|---|---|
| Standard deviation | 0.13 | 0.08 |
| Observations | 29 | 29 |
| Hypothesized mean difference | 0 | |
| df | 45 | |
| t Stat | −73.63 | |
| P(T ≤ t) one-tail | $7.45 \times 10^{-3}$ | |
| t Critical one-tail | 1.68 | |
| P(T ≤ t) two-tail | $1.49 \times 10^{-3}$ | |
| t critical two-tail | 2.01 | |

**Table A3.** *t*-test: two-sample assuming unequal variances between basic SVM and SVM-SP, simple beam.

| | **Basic SVM** | **SVM-SP** |
|---|---|---|
| Standard deviation | 0.13 | 0.08 |
| Observations | 29 | 29 |
| Hypothesized mean difference | 0 | |
| df | 49 | |
| t Stat | −58.70 | |
| P(T ≤ t) one-tail | $2.25 \times 10^{-3}$ | |
| t Critical one-tail | 1.68 | |
| P(T ≤ t) two-tail | $4.51 \times 10^{-3}$ | |
| t critical two-tail | 2.01 | |

**Table A4.** *t*-test: two-sample assuming unequal variances between basic SVM and SVM-EN, simple beam.

| | **Basic SVM** | **SVM-EN** |
|---|---|---|
| Standard deviation | 0.13 | 0.11 |
| Observations | 29 | 29 |
| Hypothesized mean difference | 0 | |
| df | 55 | |
| t Stat | −75.20 | |

**Table A4.** *Cont.*

|  | **Basic SVM** | **SVM-EN** |
|---|---|---|
| P(T ≤ t) one-tail | $1.92 \times 10^{-3}$ | |
| t Critical one-tail | 1.67 | |
| P(T ≤ t) two-tail | $3.85 \times 10^{-3}$ | |
| t critical two-tail | 2.00 | |

**Table A5.** *t*-Test: two-sample assuming unequal variances between basic SVM and SVM-MD, RC beam.

|  | **Basic SVM** | **SVM-MD** |
|---|---|---|
| Standard deviation | 0.24 | 0.22 |
| Observations | 29 | 29 |
| Hypothesized mean difference | 0 | |
| df | 35 | |
| t Stat | −20.37 | |
| P(T ≤ t) one-tail | $2.72 \times 10^{-3}$ | |
| t Critical one-tail | 1.69 | |
| P(T ≤ t) two-tail | $5.44 \times 10^{-3}$ | |
| t critical two-tail | 2.03 | |

**Table A6.** *t*-Test: two-sample assuming unequal variances between basic SVM and SVM-S2, RC beam.

|  | **Basic SVM** | **SVM-S2** |
|---|---|---|
| Standard deviation | 0.24 | 0.07 |
| Observations | 29 | 29 |
| Hypothesized mean difference | 0 | |
| df | 35 | |
| t Stat | −13.75 | |
| P(T ≤ t) one-tail | $5.57 \times 10^{-3}$ | |
| t Critical one-tail | 1.69 | |
| P(T ≤ t) two-tail | $1.11 \times 10^{-3}$ | |
| t critical two-tail | 2.03 | |

**Table A7.** *t*-Test: two-sample assuming unequal variances between basic SVM and SVM-SP, RC beam.

|  | **Basic SVM** | **SVM-SP** |
|---|---|---|
| Standard deviation | 0.24 | 0.08 |
| Observations | 29 | 29 |
| Hypothesized mean difference | 0 | |
| df | 56 | |
| t Stat | −7.65 | |
| P(T ≤ t) one-tail | $1.44 \times 10^{-3}$ | |
| t Critical one-tail | 1.67 | |
| P(T ≤ t) two-tail | $2.87 \times 10^{-3}$ | |
| t critical two-tail | 2.00 | |

**Table A8.** *t*-Test: two-sample assuming unequal variances between basic SVM and SVM-EN, RC beam.

|  | **Basic SVM** | **SVM-EN** |
|---|---|---|
| Standard deviation | 0.24 | 0.15 |
| Observations | 29 | 29 |
| Hypothesized mean difference | 0 | |
| df | 42 | |
| t Stat | −21.78 | |
| P(T ≤ t) one-tail | $8.35 \times 10^{-3}$ | |
| t Critical one-tail | 1.68 | |
| P(T ≤ t) two-tail | $1.67 \times 10^{-3}$ | |
| t critical two-tail | 2.01 | |

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
