# Peer review of "Proposed Machine Learning Techniques for Bridge Structural Health Monitoring: A Laboratory Study"

_remotesensing, doi:10.3390/rs15081984_

Round 1

Reviewer 1 Report (Previous Reviewer 1)

All the minor comments previously made by this reviewer about the paper are properly addressed. Publication is recommended.

Author Response

Dear Reviewer,

Thanks for your comment and the confirmation.

Regards.

Reviewer 2 Report (Previous Reviewer 2)

The paper is about Bridge Structural Health Monitoring using ML methods. The authors state in the introduction to develop 4 SVM models with the help of which they can solve the problem 'better'. I find this a bit misleading, because the SVM methods are already there and are only applied here specifically to the problem, but without special justification why exactly these methods or the extensions are better suited here. This is concluded exclusively from the performance. In itself, the approach in the paper is legitimate, but the focus should not be placed on the SVM with its variants, since this has already been done extensively in other papers. But more on the feature extraction for exactly this problem, because this is problem specific, but not the SVM models.

If the focus is to continue on the models and their specification, several questions arise that the authors should answer:
- can the method of SVM with misclassified always be used, what are the conditions, why does this work, and what if the problem/misclassified are very unbalanced,
- why is the sigmoid kernel appropriate here (i.e., content not outcome oriented).
- the same is true for the hybrid approach

I am missing in the paper, the rationale for the choice of methods and for the algorithms themselves.

Experimental section:

- what is meant with traditional SVM? How the paramtere are optimized? A detailed description is here missing ==> a SVM with bad kernel paramtere choice performe worse, so the authors must guarantee that the comparison is fair

Author Response

Respond to the Comments:

We sincerely thank you for the comments. Please find our answers below:
Please provide a more in-depth justification for methods selection as well as a more comprehensive 
discussion of results vs the actual focus of the paper.
 -More in-depth justification for methods selection:
To respond to this comment, comprehensive discussion has been added to the paper for justification 
of methods, lines 123-139. Also, additional results and analysis have been added, 547-560. 
-More comprehensive discussion of results:
To respond to this comment, more discussion of results has been added to the paper, and additional 
results and analysis have been added to make it comprehensive, lines 493-495, 504, 507, 532-540, 
547-560.
-Actual focus of the paper: 
To respond to this comment, the new content has been added to lines 57-63. The main contribution 
in line 67 has been re-rewritten as well. 
Another major point that must be addressed is about addressing the paper title and refocussing the 
abstract, i.e., by including a clear indication that this study has been exclusively validated at the lab
level on concrete beams. As of now, the title is misleading - as it refers to bridge structures in general 
and this aspect is also neglected in the abstract. A possible suggestion re: the title, could be
as follows: "Developed Machine Learning Techniques for Bridge Structural Health Monitoring: a 
Laboratory Study". Please also adjust/amend the abstract accordingly.
To respond to this comment, the title has changed as suggested (lines 2-3) and the abstract adjusted 
accordingly (lines 18-19).
Also, throughout the paper the word ‘developed’ replaced by ‘enhanced’ or ‘proposed’ words.
‘Traditional’ SVM replaced by ‘Basic’ SVM

Round 2

Reviewer 2 Report (Previous Reviewer 2)

Thank you very much for considering the criticism.

Unfortunately, there are still some points that should be revised:
- it should be made clear that the adapted methods are already known and are specified here only for a concrete problem.
- an explanation why this specification is useful here (I don't mean that it already worked for others)
- a clarification about the significance of the improvements in the experiments and an exact description of the parameters (which parameters were used for the BasicSVM and how were the optimal parameters determined for all methods).
- what are the limitations/ possibilities of the SVM-MD?

The formatting and layout should be improved (quality of images, adequate formulas and so on).

Author Response

We sincerely thank you for the comments. Please find our answers below:

- it should be made clear that the adapted methods are already known and are specified here only for a concrete problem.

Thank you for your comment. The complete and clear description about this comment can be found in lines 59-64.

- an explanation why this specification is useful here (I don't mean that it already worked for others)

Thank you for your comment. The complete and clear description about this comment can be found in lines 130-138.

- a clarification about the significance of the improvements in the experiments and an exact description of the parameters (which parameters were used for the BasicSVM and how were the optimal parameters determined for all methods).

Thank you for your comment. To respond to the reviewer’s comment for the first part, lines 521-525 have been added to the paper.

To respond to the reviewer’s comment for the second part, lines 489, 490, 491, 494, 497, 499 and 500 have been added to the paper.

- what are the limitations/ possibilities of the SVM-MD?

To respond to the reviewer’s comment for the second part, lines 287-290 have been added to the paper.

This manuscript is a resubmission of an earlier submission. The following is a list of the peer review reports and author responses from that submission.

Round 1

Reviewer 1 Report

This reviewer would like to congratulate the authors for their work.  The paper is within the scope of the journal, and especially within the scope of the special issue. The references are complete and up to date. The paper, that describes and compares four SVM-based approaches developed to enhance the performance of SVM for bridge structural health monitoring, is of interest to SHM specialists. The paper is well written and can be followed without much difficulty.

This reviewer recommends publication provided the following minor comments are adequately addressed: 
- Line 350: A reference to the Matlab program should be included.
- Line 364: A reference to the LabVIEW program should be included.

Reviewer 1 Report

- We sincerely thank you for the comments. Please find our answers below.

- Line 350: A reference to the Matlab program should be included.

It is not the program from other sources to reference, and it has been written by us, the authors.

- Line 364: A reference to the LabVIEW program should be included.

It is not the program from other sources to reference, and it has been written by us, the authors.

Reviewer 2 Report

In the paper the SVM and variations of it is used to predict a simple
and a RC beam.

The authors make an extensive literature research to
underline that SVM can/should be used for classification task in the
field of bridge monitoring. Yet, here I do not share the same meaning
like the authors. As I understand, the given tasks are classical
classification problems. The SVM is a pure classification scheme and
there are lot of comparisons between SVM and other classifiers also
for other standards classification applications. In this field is
nothing special, beside a task-depending feature extraction, but this
is not connected to the SVM. Thus, the content of chapter 2 "Literature Review" to not fit to the paper. The extensive Table 1 with
inconsistent descriptions has little to do with the rest of the
paper.  Instead I missed a review of the state-of the art SVM
variants, to compare these with the four 'new' SVM-models.

I also doubt a bit that these 4 SVM variants are **new** models:

  • SVM-MD: the idea is quite ok, but I missed the comparison to the state of the art and it is hard to follow the authors here. There are
    similar ideas, the authors should differentiate their idea from other
    SVM methods with similar ideas.
  •  the 2nd  'new' SVM-EN: As teh authors mention, that this idea is already known. Here they apply the idea in their scenario.
  • The same is valid for the 3rd and 4th variant.(SVM.S2 and SVM.SP).
     It is valid to apply these ideas in the field, but these
    models/ideas are not **new**.

I also missed the connection between these 3 ideas and why the authors not combined the ideas (which would be possible).  Thus, the
 focus of the paper is not clear for me. The application also comes a
 little too short, too.

Further comments to the paper:

  • The SVM itself is not introduced in a adequate way, the chapter 3.1
    and 3.2 should be reviewed by the authors again intensively.
  • a lot of abbreviation were used and not or later introduced
  • Table 2: inconsistent writing
  • the text has redundant parts
  • T-test in experimental section: the authors missed to provide the standard deviation of the accuracies, which is necessary for such tests 

Reviewer 2 Report

- We sincerely thank you for the comments. Please find our answers below.

  • The content of chapter 2 "Literature Review" to not fit to the paper. The extensive Table 1 with inconsistent descriptions has little to do with the rest of the

The proposed models are the outcome of recognizing the gaps through this literature review. In this literature review, firstly we identify the more suitable ML technique (SVM) for this area of research. Based on the literature, SVM is outperforming the most of other techniques. After identifying SVM as an effective technique in this regard, we try to identify the gaps and best techniques to develop the SVM models. The conclusion part of the papers in table 1 is highlighting the main points we have taken form the papers.

For example, “Hybrid kernels can be helpful in enhancing the accuracy, no investigation has been provided on performance of Sigmoid kernel alone in this area or combined with any other kernels, based on the knowledge of the author no or few investigations have been done on the applicability of a sigmoid kernel for damage detection in civil area.” Therefore, two of the models that we developed is using the hybrid kernels (SVM-S2 and SVM-SP). This literature also helped us to identify which kernels is beneficial to be used in this regard.

In general, this literature is highlighting the main points which we need to use for developing our models.

To address reviewer’s comment, the references [14,15,17,20,21,22,32] were deleted from table 1 and have been considered for in-text citations, and in general, literature review revised, lines 80-115.

  • I also doubt a bit that these 4 SVM variants are **new** models:

These models are new models which have been developed during four years as the part of PhD thesis.

SVM-MD: The difference with the traditional SVM have been described in lines 206-210, 213, 229-231.

SVM-S2 and SVM-SP: The concept of hybrid kernels for ML-based algorithms is not an innovation for this paper, as we mentioned in the paper, and it is the general concept (although it has been less explored in the area). The main point here is to develop the models based on new hybrid kernels that can outperform traditional SVM. Equations (21-23) corresponding to Equations (18-20) are demonstrating the developed hybrid kernels in this paper.

Corresponding to reviewer’s comment on this, I now added some description in this regard to the paper to make it more clarify, lines 261-264.

  • I also missed the connection between these 3 ideas and why the authors not combined the ideas (which would be possible).

I think the reviewer suggestion is great and something that we may consider for our future study.

However, the purpose of this work is to develop different ML-based models for the area of research. The proposed models in this paper (multiple models) will firstly help experts to investigate the different models based on their application. Secondly, it will be a good source for other researchers to develop  their ideas and suggestions on our techniques to make it more efficient for different applications, which highlights more directions for future study. For example, the reviewer suggesting the combination of all techniques in previous comments, which is one great direction for future study.

Corresponding to reviewer’s comment on this, I have added description in conclusion to make it clearer, lines 507, 521-524.

  • A lot of abbreviation were used and not or later introduced

The following abbreviations now been added and introduced:

Line 55, Smart Aggregate (SA)

Line 76, Artificial Neural Network (ANN)

Line 95, Artificial intelligence (AI)

Table 1, Auto-regression (AR), Magnetorheological damper (MR), Radial basis function (RBF)

  • Table 2: inconsistent writing

It has been revised.

  • T-test in experimental section: the authors missed to provide the standard deviation of the accuracies, which is necessary for such tests.

Additional info has been added in Appendix A, as suggested, lines 526-533.

Reviewer 3 Report

The author has read a lot of literature and propose four classification approaches (including SVM-MD, SVM-S2, SVM-SP and SVM-EN) to SHM which uses the concepts of Support Vector Machine (SVM) algorithm. It’s a very meaningful study that improves the accuracy of bridge structure health monitoring. However, there are still some questions in the article that need to be improved.

  1. Literature review section: References [18-23] have described the advantages of machine learning methods in different research fields in the form of text. Does the author consider deleting this part of the references from Table 1 to avoid duplication?
  2. Lines 250-251, the exploring of misclassified data usedequations 4-15. Equations 16-20 describes a new kernel function formula based on Polynomial and Sigmoid kernels. Please confirm whether the expression is correct.
  3. It is necessary to mark the names and positions of reinforced concrete, cracks, and instruments in Figure 5. It is difficult to understand the figure without any marks.
  4. Where do the concrete and reinforced concrete test samples used in the experiment come from (different bridge projects)?Please add an overview map of sampling points.
  5. The author should explain the meaning of the feature vectors extracted in Table 4 and it how to indicate whether the bridge structure is healthy(is there a clear value or trend indicating that the bridge structure has cracks or no cracks)?
  6. Line 412, the table corresponding to Table 4 of the text description should be Table 5.
  7. Table 5 needs to meet the production requirements of the three-line table.
  8. The author should explain the source and purpose of the data in the article, how much data is used for testing and calibration in the newly developed algorithm.
  9. Please describe the source of the real structural status and predicted structural crack classification data in Figure 6? it’s not mentioned above. In addition, the author should elaborate on the meaning of the different colored squares in Figure 6 in the title of the figure.
  10. The author should describe in detail the processing flow of the new method in the article, and conduct strict statistics and verification of the obtained results based on the case. In this research, the author focuses on the research progress of machine learning methods for bridge structure safety monitoring and the description of the four machine learning algorithms proposed in the article. The content of experimental analysis is very little.
  11. As the author mentioned in the introduction, he health monitoring of bridge structures through manual and visual inspection is expensive and time-consuming. How the four methods proposed in the article realize the health monitoring of bridge structures in the region.

Reviewer 3 Report

- We sincerely thank you for the comments. Please find our answers below.

  • Literature review section: References [18-23] have described the advantages of machine learning methods in different research fields in the form of text. Does the author consider deleting this part of the references from Table 1 to avoid duplication?

The references [14,15,17,20,21,22,32] were deleted from table 1 and have been considered for for in-text citations, and in general, literature review revised, lines 80-115.

  • Lines 250-251, the exploring of misclassified data used equations 4-15. Equations 16-20 describes a new kernel function formula based on Polynomial and Sigmoid kernels. Please confirm whether the expression is correct.

It has now been revised, lines 251,253, and 258.

  • It is necessary to mark the names and positions of reinforced concrete, cracks, and instruments in Figure 5. It is difficult to understand the figure without any marks.

The marks have been added to the image. Additional images have been added for more clarity, lines 374-381.

  • Where do the concrete and reinforced concrete test samples used in the experiment come from (different bridge projects)? Please add an overview map of sampling points.

Additional information has been provided, lines 365-381, 388-406.

  • The author should explain the meaning of the feature vectors extracted in Table 4 and it how to indicate whether the bridge structure is healthy (is there a clear value or trend indicating that the bridge structure has cracks or no cracks)?

It has now been added into Table 4. I also have added some more explanation to make it more clarify, lines 427-432.

  • Line 412, the table corresponding to Table 4 of the text description should be Table 5.

It has now been revised and changed to 5.

  • Table 5 needs to meet the production requirements of the three-line table.

The table has now been reorganized to meet the requirements. Table 6 and 7 also changed accordingly.

  • The author should explain the source and purpose of the data in the article, how much data is used for testing and calibration in the newly developed algorithm.

Additional information has been provided, lines 365-381, 388-406.

  • Please describe the source of the real structural status and predicted structural crack classification data in Figure 6? it’s not mentioned above. In addition, the author should elaborate on the meaning of the different colored squares in Figure 6 in the title of the figure.

The additional explanations have been added to line 473-481.

  • The author should describe in detail the processing flow of the new method in the article, and conduct strict statistics and verification of the obtained results based on the case. In this research, the author focuses on the research progress of machine learning methods for bridge structure safety monitoring and the description of the four machine learning algorithms proposed in the article. The content of experimental analysis is very little.

Additional analytical results have been added to the Appendix A, lines 526-533.

Some contents have been added in analysis section to make it clearer, line 429-432, 439.

The details of the methods are in lines 196-340. The analysis if these methods are based on two metric, Accuracy and F-score, which are the reliable metrics for Machine learning algorithms, lines 450-452. These results have been computed through the average of 4-fold cross validation for 100 runs and performed for 29 observations. It means we have run the program 29 times and in each time of running, it has been run 100 times with different training and testing samples which is automatically and randomly being selected from the poll of data. Then, the average of thesis 100 times is representing on of the observations. As mentioned above this process is performed 29 times. It assures of the accuracy of the techniques, lines 437-439.

In addition, the confusion matrix, lines 473-494, is an important, strict and reliable technique which shows summary of prediction results based on the extracted features form recorded signals. I have added some more explanations about the confusion matrix to make a result clearer and demonstrating how it has responded to our techniques.

Also, the T-Test we have used is showing how our results are significant in comparison to traditional techniques, lines 465-468.

Additional analytics has been added in Appendix A, lines 526-532.

  • As the author mentioned in the introduction, the health monitoring of bridge structures through manual and visual inspection is expensive and time-consuming. How the four methods proposed in the article realize the health monitoring of bridge structures in the region.

In this paper, we are not going to the concepts of IOT or how they can inform the centre regarding any defect. The sensors will be installed on the structure and the signals received by the sensors are recorded continuously. Based on proposed ML-based techniques we can identify occurrence of crack. Since the value of features which have been extracted from the sensors are different with the health state, the methods will identify this change as the crack identification.

I have added some more explanation in lines 429-432 to make it more clarify.

Round 2

Reviewer 3 Report

The authors have considered all comments, and the paper can be published.